# Understanding the Role of the Gut Microbiome and Microbial Metabolites in Non-Alcoholic Fatty Liver Disease: Current Evidence and Perspectives

**DOI:** 10.3390/biom12010056

**Published:** 2021-12-31

**Authors:** Natalia Vallianou, Gerasimos Socrates Christodoulatos, Irene Karampela, Dimitrios Tsilingiris, Faidon Magkos, Theodora Stratigou, Dimitris Kounatidis, Maria Dalamaga

**Affiliations:** 1Department of Internal Medicine, Evaggelismos General Hospital, 45-47 Ypsilantou Street, 10676 Athens, Greece; dimitriskounatidis82@outlook.com; 2Department of Biological Chemistry, Medical School, National and Kapodistrian University of Athens, 75 Mikras Asias, Goudi, 11527 Athens, Greece; gerchristod82@hotmail.com (G.S.C.); madalamaga@med.uoa.gr (M.D.); 32nd Department of Critical Care, Medical School, University of Athens, Attikon General University Hospital, 1 Rimini Street, Chaidari, 12462 Athens, Greece; eikaras1@gmail.com; 4First Department of Propaedeutic Internal Medicine, School of Medicine, National and Kapodistrian University of Athens, Laiko General Hospital, 17 St Thomas Street, 11527 Athens, Greece; tsilingirisd@gmail.com; 5Department of Nutrition, Exercise, and Sports, University of Copenhagen, 2000 Frederiksberg, Denmark; fma@nexs.ku.dk; 6Department of Endocrinology and Metabolism, Evaggelismos General Hospital, 45-47 Ypsilantou Street, 10676 Athens, Greece; theodorastratigou@yahoo.gr

**Keywords:** microbiota, multi-omics, NAFLD, NASH, probiotics, prebiotics, bacteriophage

## Abstract

Non-alcoholic fatty liver disease (NAFLD) is the most common chronic liver disease worldwide. NAFLD begins as a relatively benign hepatic steatosis which can evolve to non-alcoholic steatohepatitis (NASH); the risk of cirrhosis and hepatocellular carcinoma (HCC) increases when fibrosis is present. NAFLD represents a complex process implicating numerous factors—genetic, metabolic, and dietary—intertwined in a multi-hit etiopathogenetic model. Recent data have highlighted the role of gut dysbiosis, which may render the bowel more permeable, leading to increased free fatty acid absorption, bacterial migration, and a parallel release of toxic bacterial products, lipopolysaccharide (LPS), and proinflammatory cytokines that initiate and sustain inflammation. Although gut dysbiosis is present in each disease stage, there is currently no single microbial signature to distinguish or predict which patients will evolve from NAFLD to NASH and HCC. Using 16S rRNA sequencing, the majority of patients with NAFLD/NASH exhibit increased numbers of Bacteroidetes and differences in the presence of Firmicutes, resulting in a decreased F/B ratio in most studies. They also present an increased proportion of species belonging to *Clostridium*, *Anaerobacter*, *Streptococcus*, *Escherichia*, and *Lactobacillus*, whereas *Oscillibacter*, *Flavonifaractor*, *Odoribacter*, and *Alistipes* spp. are less prominent. In comparison to healthy controls, patients with NASH show a higher abundance of Proteobacteria, Enterobacteriaceae, and *Escherichia* spp., while *Faecalibacterium prausnitzii* and *Akkermansia muciniphila* are diminished. Children with NAFLD/NASH have a decreased proportion of *Oscillospira* spp. accompanied by an elevated proportion of *Dorea*, *Blautia*, *Prevotella copri*, and *Ruminococcus* spp. Gut microbiota composition may vary between population groups and different stages of NAFLD, making any conclusive or causative claims about gut microbiota profiles in NAFLD patients challenging. Moreover, various metabolites may be involved in the pathogenesis of NAFLD, such as short-chain fatty acids, lipopolysaccharide, bile acids, choline and trimethylamine-N-oxide, and ammonia. In this review, we summarize the role of the gut microbiome and metabolites in NAFLD pathogenesis, and we discuss potential preventive and therapeutic interventions related to the gut microbiome, such as the administration of probiotics, prebiotics, synbiotics, antibiotics, and bacteriophages, as well as the contribution of bariatric surgery and fecal microbiota transplantation in the therapeutic armamentarium against NAFLD. Larger and longer-term prospective studies, including well-defined cohorts as well as a multi-omics approach, are required to better identify the associations between the gut microbiome, microbial metabolites, and NAFLD occurrence and progression.

## 1. Introduction

Non-alcoholic fatty liver disease, also known as NAFLD, affects approximately 80–100 million people or approximately 25% of the total adult population in the United States. NAFLD is currently the most common cause of chronic liver disease worldwide [1,2]. It is defined as liver steatosis, i.e., an accumulation of fat in the liver exceeding 5% of the liver’s total weight, in the absence of significant alcohol consumption [2]. The overall global prevalence of NAFLD revealed by abdominal imaging is estimated at 25%, with the lowest prevalence in Africa (13.5%) and the highest in the Middle East (31.8%) [2]. About 30% of patients with NAFLD progress to non-alcoholic steatohepatitis (NASH), which is characterized by steatosis with the addition of infiltration by inflammatory cells and different stages of fibrosis (F), ranging from F0 (no fibrosis) to F4 (cirrhosis) [2,3]. The overall prevalence of NASH is approximately between 1.5–6.5% in the US adult population [2]. NASH may reverse to simple steatosis or may worsen to cirrhosis or even hepatocellular carcinoma (HCC) [3,4]. NAFLD progression and staging are depicted in Figure 1. HCC constitutes the fifth most common cancer in men and the second cause of cancer-related deaths worldwide [5,6]. HCC has an estimated incidence of 1–2% per year among patients with NASH and cirrhosis [6,7]. Moreover, NAFLD may increase cardiovascular risk, being also linked to higher rates of chronic kidney disease and its progression [1,2,3]. Besides adults, NAFLD and NASH are also rising among adolescents, in parallel with obesity, and are expected to haunt the forthcoming generations in the future [8,9].

NAFLD and NASH have been associated with a plethora of metabolic risk factors, such as overweight/obesity, type 2 diabetes mellitus (T2DM), prediabetes, hypertension, and dyslipidemia [2]. Lately, expert panels have proposed a change in the nomenclature of NAFLD, which overemphasizes the absence of alcohol and underemphasizes the role of multiple metabolic factors, to metabolic (dysfunction)-associated fatty liver disease (MAFLD) [10]. MAFLD is characterized by hepatic steatosis combined with the presence of overweight/obesity or T2DM or at least two of the following metabolic risk factors: (1) waist circumference ≥102 cm in men or ≥88 cm in women or ≥90/80 among those of Asian descent; (2) triglyceride (TG) concentration ≥150 mg/dL or treatment for hypertriglyceridemia; (3) high-density lipoprotein cholesterol concentration <40 mg/dL in men and <50 mg/dL in women or medication for dyslipidemia; (4) systolic blood pressure ≥130 mm Hg or diastolic pressure ≥85 mmHg or treatment for arterial hypertension; (5) homeostasis model assessment of insulin resistance (HOMA-IR) ≥2.5; and (6) C-reactive protein concentration >2 mg/L [10]. Although changes in the nomenclature may mirror the pathophysiology of this disorder, they could cause ambiguity as NAFLD is considered a heterogeneous disorder not invariably related to the presence of metabolic syndrome [1]. Interestingly, NAFLD may also be present in individuals without obesity [11]. Indeed, in a recent meta-analysis, around 40% of the global NAFLD population was categorized as non-obese and almost a fifth was lean (normal body weight) [11].

The prevalence and severity of NAFLD increases with age, reaching a peak at the ages between 45 and 64 years. NAFLD is more frequent in men than women in Caucasian subjects [11]. NAFLD and NASH are clinical entities that have a genetic predisposition and epigenetic components [12,13]. The pathogenesis of NAFLD represents a complex process implicating numerous factors—genetic, metabolic, and dietary—intertwined in a multi-hit etiopathogenetic model, as shown in Figure 1. Several studies have pointed out the role of specific genes, such as patatin-like phospholipase containing 3 gene (*PNPA3)*, farnesyl-diphosphate farnesyl transferase 1 gene (*FDFT1)*, transmembrane 6 superfamily protein 2 gene (*TM6SF2)*, glucokinase regulator gene (*GCKR)*, and membrane bound O acyltransferase domain containing 7 gene (*MBOAT7)*, in the development and progression of NAFLD to NASH and HCC [12,13,14,15,16,17,18]. Furthermore, the familial clustering of cases of NAFLD has been reported [19,20]. Numerous environmental factors, such as high-fat-diets, diets rich in fructose- and simple-sugar-containing beverages, and diets low in omega-3 and omega-6 unsaturated fatty acids, in conjunction with a sedentary lifestyle and a low level of physical activity, have been implicated in the development and progression of NAFLD [21,22,23,24,25,26]. Recent data have also highlighted the role of the gut metagenome in the etiopathogenesis of NAFLD.

The human microbiome comprises the sum of each and every gene from the bacteria, archaea, viruses, and eukaryotic microbes that inhabit the human body, most of which reside in the gut. The adult gut bacteria belong mainly to two phyla, the Gram-positive Firmicutes and the Gram-negative Bacteroidetes, while Actinobacteria, Proteobacteria, Fusobacteria, and Verrucomicrobia are less frequently encountered in comparison with Firmicutes and Bacteroidetes [27]. There is accumulating evidence that the gut microbiota plays a key role in physiological homeostasis by coordinating immune system reactions by means of modulating the microenvironment in the gut. However, changes in the gut microbiota due to genetic or environmental factors, such as nutritional features or medications, e.g., antibiotics or non-steroid anti-inflammatory drugs, may result in the modulation of the structure or diversity of the gut microbiota, known as dysbiosis [27,28]. In turn, dysbiosis may lead to metabolic derangement, such as T2DM, obesity, the metabolic syndrome, and NAFLD [28,29].

The aim of the present review is to: (1) summarize the role of the gut microbiome in NAFLD pathogenesis; (2) shed light on the distinct microorganisms that seem to be predominant in NAFLD and their metabolic signatures; and (3) provide insight into the potential preventive and therapeutic interventions related to gut microbiota, such as the administration of probiotics, especially next-generation probiotics, e.g., *Akkermansia muciniphila* and *Faecalibacterium prausnitzii*, prebiotics, or synbiotics, as well as highlight the contribution of bariatric surgery, bacteriophages, and fecal microbiota transplantation (FMT) in the therapeutic armamentarium against NAFLD.

## 2. NAFLD, Gut Dysbiosis, and Microbial Signatures

NAFLD is characterized by the accumulation of fat in the form of TG in hepatocytes. Hepatic TGs are formed from the esterification of fatty acids in the liver. The main sources of fatty acids for the liver are the systemic plasma free fatty acids (FFAs), originating from the lipolysis of the adipose tissue TG, and fatty acids synthesized de novo in the liver from simpler precursors, e.g., carbohydrates (lipogenesis) [30]. Gut dysbiosis, which refers to translocation, may render the bowel more permeable, leading to an increased fatty acid absorption. This increased gut permeability may result in bacterial migration via the gut epithelial barrier, with a parallel release of toxic bacterial products, lipopolysaccharides (LPS), and proinflammatory cytokines, which can initiate and sustain inflammation. This process is facilitated by the activation of Nuclear-Factor-kappa-B (NF-κB) through the Toll-like receptor 4 (TLR-4) in the host cells. Moreover, the stimulation of TLR4 may induce changes in cellular metabolism associated with fatty-acid-activated inflammation. The hepatic tissue is very sensitive to this process as it filters out a significant portion of the blood coming in through the portal vein (gut–liver axis). Intestinal microbiota could also alter bile acid metabolism, contributing to the pathogenesis of NAFLD by modulating farnesoid X receptor (FXR) stimulation and thus affecting fat and glucose homeostasis [31].

Patients with NAFLD and especially NASH have been shown to exhibit an increased number of Bacteroidetes and differences in the presence of Firmicutes, resulting in a decreased F/B ratio in most studies [9]. Notably, the F/B ratio may vary and not yield consistent results in all studies as it highly depends on the molecular methods used to identify the bacteria, i.e., 16S rRNA versus shotgun metagenome sequencing. Moreover, there are huge diversities in the microorganisms within each of these phyla, which renders the F/B ratio a rather crude estimate. Furthermore, these disturbances in the F/B ratio have not been documented among patients with HCC. Apart from this difference, patients with NAFLD have also been demonstrated to exhibit an increased proportion of species belonging to *Clostridium*, *Anaerobacter*, *Streptococcus*, *Escherichia*, and *Lactobacillus*, whereas *Oscillibacter*, *Flavonifaractor*, *Odoribacter*, and *Alistipes* spp. are less prominent [32]. Besides, there is a relative abundance of potential pathogens, such as Gram-negative Proteobacteria, Enterobacteriaceae, and *Escherichia* spp. among patients with NASH, when compared to healthy controls, while *Faecalibacterium prausnitzii* and *Akkermansia muciniphila* are relatively diminished [33,34,35]. *Faecalibacterium prausnitzii*, a Gram-positive anaerobe bacterium, *Eubacterium rectale*, *Eubacterium halii*, and *Anaerostipes caccae* are well-known short-chain fatty acids (SCFAs) producers, particularly butyrate. *Akkermansia muciniphila*, a mucin-producing, Gram-negative anaerobe bacterium, when co-cultured with the butyrate-producing *Faecalibacterium prausnitzii*, *Eubacterium rectale*, *Eubacterium halii*, and *Anaerostipes caccae* results in an enhanced production of butyrate. Therefore, *Akkermansia muciniphila*, apart from its beneficial role in the gut epithelium per se, may promote the growth of other bacteria with anti-inflammatory properties [33,34,35]. Children with NAFLD and NASH have a decreased proportion of *Oscillospira* spp. accompanied by an elevated proportion of *Dorea*, *Blautia*, *Prevotella copri*, and *Ruminococcus* spp. when compared to healthy controls [36]. Figure 2 shows the altered gut microbiota found in patients with NAFLD/NASH. Table 1 depicts major studies in animal models, while Table 2 portrays major studies in humans regarding the gut microbiota signatures in NAFLD.

The alteration in the gut microbiota is related to higher fecal concentrations of 2-butanone and 4-methyl-2-pentanone, metabolites known to cause hepatocellular toxicity in individuals with metabolic liver diseases, when compared to healthy individuals [37]. In addition, due to the fact that the gut microbiota in patients with NALFD is enriched in ethanol-producing bacteria, such as *E. coli*, which is capable of producing ethanol under anaerobe conditions, it has been suggested that this rich gut microbiota may produce more ethanol than the microbiota of healthy individuals, as evidenced by the increased concentrations of intrinsically generated ethanol in the circulation as well as in the breath [33,37]. Ethanol is known to stimulate NF-κB signaling molecules to provoke tissue damage, via impairing gut barrier function and, thus, contributing to increased portal LPS concentrations [38,39]. It has been documented that the detoxification pathway is weakened in the liver of patients with NALFD, resulting in an increased production of reactive oxygen species (ROS), which have the potential to cause oxidative damage to the hepatocytes, resulting in augmented hepatic inflammation and subsequently contributing to NASH [40]. Results from the scarce human and animal fecal transplantation studies found a higher abundance of the alcohol-producing bacterial species *Klebsiella pneumoniae* in the gut, which led to acceleration in the pathogenesis of NAFLD [41,42,43].

**Table 1 biomolecules-12-00056-t001:** Differences in microbial species abundance in various animal models.

Animal Studies		
Study, Year	Animal Model	Remarks
Rahman et al., 2016 [44]	Knockout mice of the F11 receptor gene, a gene conferring a junctional adhesion molecule A, implicated in derangement in intestinal permeability	*↑* Firmicutes *↑* Proteobacteria
Pierantonelli et al., 2017 [45]	NLRP3 Knockout mice	↓ Gram negative species ↓ Bacterial translocation after treatment with antibiotics
Llorente et al., 2017 [46]	Sublytic Atp4aSl/Sl mice treated with PPIs	*↑**Enterococcus faecalis* with PPIs
Gart et al., 2018 [47]	Leiden mice	Variations in gut microbiota, non-specific
Schneider et al., 2019 [48]	Rats with methionine-choline deficient diet-induced NASH	↓ Gut microbiota diversity
Petrov et al., 2019 [49]	GF-HFD not responders	*↑ Desulfovibrio* *↑* *Oscillospira* *↓ Bacteroides* *↓ Oribacterium*
Chen et al., 2019 [50]	Knockout SIRT3 HFD mice	*↑ Desulfovibrio* *↓ Oscillibacter* *↓ Alloprevotella*
De Sant’Ana et al., 2019 [51]	Knockout mice (caspases 1/11 and NLRP3 HFD)	*↑* Proteobacteria *↑* F/B ratio
Ahmad et al., 2020 [52]	Mice C57BL/6J HFD	Alterations in Prevotellaceae UCG-003, Ruminococcaceae UCG-005, *Desulfovibrio*, the Lachnospiraceae NK4A136 group, *Lactobacillus* and *Akkermansia*
Cavallari et al., 2020 [53]	NOD2 Knockout mice	*↑* Clostridiales *↓* Erysipelotrichaceae
Zhang et al., 2021 [54]	Mice, C57BL/6 male, high-fat, high-cholesterol diet	*↑ Mucispirillum* *↑ Desulfovibrio* *↑ Anaerotruncus* *↑* Desulfovibrionaceae *↓ Bifidobacterium* *↓ Bacteroides*

Abbreviations: F/B ratio: *Firmicutes* to *Bacteroidetes* ratio; GF: Germ Free; HFCD: High-Fat, High-Cholesterol Diet; HFD|: High-Fat Diet; *↑*: increased, *↓*: decreased.

**Table 2 biomolecules-12-00056-t002:** Evidence from human studies depicting associations of various bacterial species and metabolic signatures in patients with NAFLD.

Human Studies				
Study, Year	Population	Lab Techniques	Microbiome	Remarks
Belgaumkar et al., 2016 [55]	NAFLD as described by serum cytokeratin 18, 18 patients who underwent laparoscopic sleeve gastrectomy (UK)	Serum: Liquid chromatography tandem-mass spectometry for BA	No bacteria were further detected	Total BA did not change; ↓ primary glycine- and taurine-conjugated BA, ↓ cholic acid, and ↑ secondary BA, ↑ glycine-conjugated urodeoxycholic acid over the study period. These changes are associated with reduction in insulin resistance, pro-inflammatory cytokines, and CK-18 levels
Boursier et al., 2016 [56]	Biopsy-proven NAFLD among 57 patients (France, USA)	Fecal Microbiome: 16S rRNA gene Sequencing	Patients with NASH and F2≥2: ↑ *Bacteroides* ↓ *Prevotella.* Patients with F2 ≥ 2: ↑↑ *Ruminococcus*	NASH was related to ↑ *Bacteroides*, while significant fibrosis to *↑↑* *Ruminococcus*
Loomba et al., 2017 [57]	Biopsy-proven NAFLD among 86 patients (USA)	Fecal Microbiome: Whole-genome shotgun sequencing of DNA from feces	Patients with NAFLD: *↑* Proteobacteria *↑* Firmicutes Patients with NAFLD and ≤F2: *↑ Eubacterium rectale* *↑ Bacteroides vulgatus* Patients with NAFLD and >F2: *↑ Bacteroides vulgaris* *↑ Escherischia coli*	Patients with NAFLD and ≤F2: ↑ Lactate ↑ Acetate ↑ Formate Patients with NAFLD and >F2: ↑ Butyrate ↑ D-Lactate ↑ Propionate ↑ Succinate
Del Chierico et al., 2017 [36]	NAFLD in 61 patients and 51 non-NAFLD controls (Italy)	Fecal Microbiome: rRNA Sequencing Serum metabolites: GC/MS	Patients with NAFLD: *↑* Actinobacteria *↓* Bacteroidetes *↑* *Ruminococcus* *↑ Blautia* *↑ Dorea* *↓* *Oscillospira* *↓* Rikenellaceae	Patients with NAFLD: ↑ 2-butanone ↑ 1-pentanol ↑ 4-methyl-2-pentanone
Puri et al., 2018 [58]	Biopsy-proven NAFLD among 86 patients and 24 non-NAFLD controls (USA)	Serum metabolites: LC/MS	No bacteria were further detected	Patients with NAFLD and ≥F2: ↑ conjugated cholate ↓ ratio of total secondary to primary BAs Patients with NASH had ↑↑ total conjugated primary BAs when compared to controls
Hoyles et al., 2018 [59]	Biopsy-proven NAFLD among 56 patients (UK, Italy, France)	Fecal Microbiome: Shotgun Metagenomic Sequencing Serum and urine metabolites: LC/MS	Among patients with steatosis: ↑ Proteobacteria *↑* Actinobacteria	Among patients with steatosis: -Serum BCAAs: ↑ leucine ↑ valine ↑ isoleucine ↑ phenylacetic acid -Urine: ↑ choline
Caussy et al., 2018 [60]	Discovery cohort of 156 twins Validation cohort of Biopsy-proven NAFLD among 156 patients (USA, France)	Fecal Microbiome: Whole Shotgun Metagenomics Sequencing Liver: MRI-PDFF; MRE Serum metabolites: LC/MS; GC/MS	Patients with NAFLD and >F2: *↑* Furmicutes *↑* Bacteroidetes *↑* Proteobacteria	56 metabolites had a relationship with hepatic fibrosis, among which 3-(4-hydroxyphenyl) lactate, N-formylmethionine, phenyllactate, mannitol, allantoine and N-(2-furoyl) glycine were the most abundant 3-(4-hydroxyphenyl) lactate was ↑↑ in liver fibrosis and steatosis
Caussy et al., 2019 [61]	Cross-sectional; 203 participants including NAFLD and healthy controls (USA)	Fecal Microbiome: 16S rRNA Sequencing Liver: MRI/MRE	Patients with NAFLD and cirrhosis: ↑ Enterobacteriaceae ↑ *Streptococcus* *↑ Gallibacterium* ↑ *Megasphaera* A trend towards Gram negative species in advanced fibrosis was reported	No metabolites were further detected
Lee et al., 2020 [62]	Biopsy-proven NAFLD among 171 patients and 31 non-NAFLD controls (USA, Korea)	Fecal Microbiome: 16S rRNA Sequencing	Patients with NAFLD and >F2, non-obese: ↑ Ruminococcaceae ↑ Veillonellaceae	Patients with NAFLD and >F2, non-obese: ↑ BA ↑ Propionate in feces
Adams et al., 2020 [63]	Biopsy-proven NAFLD among 67 patients and 55 non-NAFLD controls (USA)	Fecal Microbiome: 16S rRNA Sequencing Serum and fecal metabolites: HPLC/MS	Patients with NAFLD and >F2: ↑ Firmicutes ↑ Proteobacteria ↑ Actinobacteria ↓ Bacteriodetes ↑ Actinomycetaceae ↓ Lachnospiraceae	Patients with NAFLD and >F2: ↑ BA in serum and feces
Masarone et al., 2021 [64]	Biopsy-proven NAFLD among 144 patients (Italy)	Serum metabolites: GC/MS	No bacteria were further detected	Patients with NAFLD and >F2: ↑ Glycocholic acid ↑ Taurocholic acid ↑ Phenylalanine ↑ BCAAs ↓ Glutathione
Nimer et al., 2021 [65]	Biopsy-proven NAFLD among 102 patients and 50 non-NAFLD controls (USA)	Plasma BA metabolites: LC/MS	No bacteria were further detected	Patients with NAFLD and >F2: ↑↑ Plasma 7-keto-DCA levels Some glycine conjugated forms of BAs ↑↑ in more advanced stages of NAFLD

Abbreviations: BA: Bile Acids; BCAAs: Branched-Chain Amino Acids; FMT: Fecal Microbiota Transplantation; GC/MS: Gas Chromatography/Mass Spectrometry; LC/MS: Liquid Chromatography/Mass Spectrometry; MRE: Magnetic Resonance Elastography; MRI-PDFF: Magnetic Resonance Imaging Proton Density Fat Fraction; NAFLD: Non-Alcoholic Fatty Liver Disease; NASH: Non-Alcoholic Steatohepatitis; qPCR: quantitative Polymerase Chain Reaction; WGS: Whole Genome Shotgun; *↑*: increased, *↓*: decreased.

Animal studies have yielded different results regarding the microbial species in models with NAFLD. These differences may be attributed to the different animal models used, i.e., differences in the knock-out mice and deleted genes. However, most studies have documented a state of gut dysbiosis in NAFLD [44,45,46,47,48,49,50,51,52,53,54]. Overall, human studies have found differential abundances among patients with NAFLD and especially among patients with severe NAFLD associated with fibrosis—and particularly among those staged ≥F2. A variety of changes in the abundance of Bacteroidetes, Firmicutes, and especially *Ruminococcus* has been observed, with either increases or decreases in the relative abundances of the abovementioned species. These differences may be attributed to several reasons, mainly the different molecular techniques used to describe bacteria to the species level and the differences in the methodologies used for the definition of NAFLD and especially NASH. Regarding molecular techniques for the description of the gut microbiota, methods such as the Polymerase Chain Reaction (PCR) and the 16S rRNA gene amplicon sequencing and Next-Generation Sequencing (NGS), such as the shotgun metagenome sequencing, have shed light on the abundance of different species in the gut microbiota of patients with NAFLD/NASH/HCC. However, it is exactly the advent of differences in the methodologies used for DNA extraction, PCR, and NGS techniques which may contribute to the variability in the relative abundances of the different species found. In addition, the site from which the specimen has been acquired, e.g., rectum or caecum, as well as the type of specimen, e.g., feces versus biopsy specimen, may account for the reported differences in the isolated microbiota species. Regarding the diagnosis and staging of NAFLD/NASH, the gold standard remains the liver biopsy. Nevertheless, a plethora of imaging techniques, such as ultrasound, contrast ultrasound, computed tomography, magnetic resonance imaging, ultrasound elastography, and magnetic resonance elastography, in conjunction with several noninvasive biomarkers, are commonly used instead of a liver biopsy due to their availability and safety. Liver biopsy is an invasive and expensive diagnostic method, critically depending on the experience of the physician and increasing the risk of complications. Therefore, human studies have the drawback of using different molecular techniques and, moreover, different diagnostic methodologies to characterize patients with NAFLD/NASH. These differences may account for the differential abundances of bacterial species of the gut microbiota among patients with NAFLD/NASH [36,55,56,57,58,59,60,61,62,63,64,65]. Further large-scale, homogeneous, and longitudinal studies are needed to further categorize the microbial signatures of patients with NAFLD/NASH.

## 3. Microbiome-Derived Compounds in the Pathogenesis of NAFLD

Various metabolites have been implicated in the pathogenesis of NAFLD, such as short-chain fatty acids, LPS, bile acids, choline and trimethylamine-N-oxide, and ammonia levels. The abovementioned substances have all been involved in the etiopathogenesis of NAFLD, as portrayed in Figure 3.

SCFAs act by promoting intestinal integrity, whereas the LPS has a negative effect on this functional barrier. Differences in bile acids may also affect the dynamics of their portal circulation, thereby influencing NAFLD development, while choline deficiency has been related to a reduced hepatic production of very-low-density lipoproteins (VLDL), resulting in the accumulation of TG within the liver, thus promoting NAFLD. Ammonia is a marker of hepatic encephalopathy but is also suggested to be involved in the pathogenetic mechanisms of NAFLD.

### 3.1. SCFAs

SCFAs, mainly acetate, propionate, and butyrate are organic fatty acids synthesized from non-digestible proteins and fibers via anaerobic fermentation by the gut microbiota [66,67]. They are mainly produced in the distal colon, where they serve as a complementary factor to intestinal integrity and function. SCFAs are transferred to the liver by means of the portal circulation, thereby serving as precursors for gluconeogenesis and lipogenesis [68,69]. In fact, SCFAs are responsible for about 5–10% of the typical energy demands under normal conditions [70,71]. There are numerous studies demonstrating that SCFAs activate the G-protein-coupled receptors (GPCRs) GPR41 and GPR43 in the surface of the gut enteroendocrine L cells. In particular, activation of the GPCRs stimulates peptide YY (PYY) release, which results in the slowing down of gastric emptying and the promotion satiety [72,73]. In addition, activation of GPR41 and GPR43 on the surface of the L cells increases secretion of GLP-1, an incretin known to slow gastric emptying and induce satiety, thus decreasing food intake. Furthermore, GLP-1 enhances lipid oxidation in the liver, which contributes to diminished steatosis [73,74,75,76]. Besides their functionality as energy substrates, SCFAs have the potential to affect hepatic metabolism by functioning as signaling molecules. In particular, propionate and butyrate activate AMP-activated protein kinase (AMPK) to promote hepatic autophagy, a catabolic process which results in the hydrolysis of TG and the release of fatty acids for β-oxidation in the mitochondria [76,77,78,79,80]. The activation of AMPK by SCFAs has been related to increased Uncoupling Protein 2 (UCP2) levels and an increased AMP:ATP ratio [81]. Apart from the AMPK activation, SCFAs inhibit class I and II histone deacetylases and can thus alter gene transcription. Class I and II histone deacetylases are enzymes which catalyze the removal of acetyl groups from lysine residues on histones to reduce gene transcription. Butyrate and propionate have been shown to inhibit histone deacetylases in human colon carcinoma cells, whilst in macrophages the inhibitory effect of butyrate on histone deacetylases is likely to be responsible for its anti-inflammatory properties [76,77,78,79,80,81]. Loomba et al. have reported that patients with advanced fibrosis have increased acetate levels in their fecal samples, whereas patients with mild or moderate NAFLD presented increased levels of butyrate and propionate [20]. However, when circulating SCFAs were measured in cirrhosis patients the serum butyric acid levels inversely correlated to the inflammatory markers and serum endotoxin levels [76,77,78,79,80]. These differences may be attributed to a plethora of factors, such as variations in age, diet, environmental parameters, or even sample handling and processing. More specifically, the estimation of serum or fecal SCFA levels is troublesome per se as SCFAs are volatile substances, requiring immediate processing for an accurate measurement. However, SCFA supplementation in mouse models of NAFLD has shown beneficial effects. In High Fat Diet (HFD)-fed rodents, supplementation with butyrate resulted in a reduction in liver and adipose tissue inflammation. Besides, butyrate promoted alterations in the bacterial population of the gut microbiota. In particular, it enhanced SCFA-producing bacteria and reduced endotoxin-releasing bacteria [82]. Based on these animal model data, SCFA supplementation may have beneficial metabolic effects as well as decrease the severity of liver steatosis. Large-scale studies of SCFA supplementation among patients with NAFLD are lacking. It would be really intriguing to investigate SCFA supplementation in RCTs in humans.

### 3.2. Endotoxins

Inflammation is a hallmark feature of NASH, in which gut microbiota play a pivotal role [83]. Bacterial products stemming from gut microbiota, such as LPS, peptidoglycan, and bacterial DNA, may be transferred via the portal vein to activate the TLRs on Kupffer cells, leading to an inflammatory cascade which promotes the development of NASH. Elevated levels of LPS have been detected in NAFLD in rodent (rats and mice) and human studies [84,85,86]. Pathogen-associated molecular pattern (PAMP) receptors, such as TLRs, are deeply implicated in the pathogenesis of NASH by the activating of NF-κB, inducing the secretion of chemokines from macrophages, and the recruiting of Kupffer cells to the liver to promote the inflammation process [84,85,86,87,88,89,90]. In addition, Nod-like receptor protein (NLRP)-3 may stimulate immunity by means of forming an inflammasome with ACS (the adaptor molecule apoptosis-associated speck-like protein containing a CARD), an apoptosis-associated protein, in order to activate pro-caspase 1 [91]. Inflammasome dysfunction leads to exaggerated liver inflammatory response, liver fibrosis, and cell death [91]. This role of NLRP3 has been documented in HFD-fed rodents, which exhibited decreased liver steatosis by inhibition of the NLRP3 inflammasome pathway [92]. Several TLRs have been shown to be of key importance, with the most significant being TLR-4 and TLR-9. For example, rodents deficient in TLR4 and myeloid-differentiation factor-2 (MD2) are protected from methionine- and choline-deficient diet-induced liver inflammation and liver steatosis [93]. Furthermore, plasma from patients with NASH has been found to possess increased levels of mitochondrial DNA as a potent TLR-9 activator. Mice deficient in TLR9 have been documented to be protected from HFD-induced liver steatosis and inflammation, thus pointing out the importance of the TLR-9 pathway in modulating inflammation in NASH [91,92,93]. Lastly, TLR-5 may play a protective role in diet-induced NASH, as mice lacking TLR-5 on hepatocytes showed exacerbated disease after being fed with a methionine- and choline-deficient diet [94]. These examples help clarify how PAMPs may provoke inflammation in the liver and suggest an interplay of bacteria and gut microbiota in the pathogenesis of NASH.

### 3.3. Bile Acids

Bile acids (BAs) are mainly produced by cholesterol in the liver. They are categorized as primary BAs, such as cholic acid (CA) and chenodeoxycholic acid (CDCA), and secondary BAs, such as deoxycholic and lithocholic acid [95,96]. Primary BAs are further conjugated with glycine or taurine and stored in the gallbladder before being released into the intestine after consumption of a meal. In the gut, BAs are implicated in the absorption of dietary fat, cholesterol, and fat-soluble vitamins [97,98]. The primary BAs are deconjugated and dehydroxylated by gut microbiota to the more hydrophobic secondary BAs, which are reabsorbed in the distal ileum and returned to the liver by means of the portal vein [97,98,99]. There are several studies in favor of the notion that there are specific BA profiles related to NASH [100,101]. For example, Yara et al. have analyzed serum BAs and have documented that the ratios of primary to secondary BAs, taurine-conjugated BAs to glycine-conjugated BAs, unconjugated BAs to total BAs, and secondary BAs to total BAs are reduced in NASH patients compared to those of healthy individuals [100]. Moreover, Chen et al. have documented that increased ratios of circulating conjugated Chenodeoxycholic acids (CDCAs) to muricholic acids in NASH patients are correlated to the histological severity of NASH and the grade of fibrosis [101]. In addition, a BA intermediate and marker for de novo BAs synthesis, 7α-hydroxy-4-cholesten-3-one (C4), has been shown to be increased in the serum of patients with NASH and has also been related to changes in the gut microbiota [102].

BAs, especially secondary ones, may serve as signaling molecules by binding to cellular receptors, such as the FXR and the G protein-coupled bile acid receptor 1 (also known as TGR5). Different BAs possess variable abilities to activate these receptors [103]. For example, secondary BAs are more potent than primary BAs in activating TGR5 [104]. TGR5 is ubiquitously expressed throughout the human body with increased levels of TGR5 mRNA detected in metabolically active organs, such as the small intestine, stomach, and liver. Activating TGR5 has been shown to increase the intestinal GLP-1 release in obese animal models [104,105]. TGR5 has also been documented to be expressed in monocytes, macrophages, and Kupffer cells, modulating immune responses [104,105,106]. Indeed, in isolated Kupffer cells, bile acids activated TGR5 and inhibited LPS-induced cytokine expression in a cAMP-dependent manner [104,105,106]. However, there is a scarcity of studies in animal models as well as in humans regarding the role of BAs in the pathogenesis of NAFLD in animal models as well as in humans.

### 3.4. Choline and TMAO

Choline is mainly obtained from dietary sources, such as red meat, eggs, cheese, and peanuts, although de novo choline synthesis may also take place in the liver [107]. Choline is a component of the cell membrane, necessary for the production of phosphatidylcholine and sphingomyelin, which are indispensable structural and functional membrane phospholipid components. In the liver, choline is also necessary for the production of VLDL. Therefore, choline deficiency may lead to a reduced production of VLDL, resulting in the accumulation of TG in the liver [108,109]. For this reason, choline-deficient diets have been used in animal models to induce NASH [110]. Choline is known to be converted to trimethylamine (TMA) by gut microbiota. TMA can be oxidized by hepatic monooxygenases to produce trimethylamine N-oxide (TMAO) in the liver, which is afterwards released in the systemic circulation [107]. In HFD-fed rodents, higher conversion of choline to TMA by microbiota resulted in lower bioavailability of choline [111]. TMAO may also act directly on the liver and contribute to impaired glucose tolerance and the development of NAFLD [7,112]. In particular, studies have demonstrated that serum levels of TMAO are higher in patients with NAFLD than in healthy controls, being also correlated with the severity of hepatic steatosis [113]. Another study reported that elevated serum TMAO levels are related to NASH in patients with T2DM [114]. TMAO levels have also been associated with atherosclerosis via the increased production of foam cells in animal models. In particular, TMAO has been demonstrated to promote macrophage migration and their transformation into foam cells, while endothelial dysfunction as well as platelets dysfunction and thrombus formation have been correlated to increased TMAO levels. Based on these mechanisms, high blood TMAO has been suggested as a contributor to the increased cardiovascular disease risk. However, it remains unknown whether serum TMAO levels may serve as a biomarker for NAFLD prognosis or other metabolic derangements or whether it may function as a potential therapeutic target for atherosclerosis [115].

### 3.5. Ammonia

Hyperammonemia is suggested to be a marker of the severity of liver disease [116]. During NASH, ornithine transcarbamylase and carbamoyl phosphate synthetase mRNA, protein, and activity have been shown to be decreased, leading to hyperammonemia [117]. Notably, ammonia itself has been suggested to exert direct effects on hepatic stellate cells by activating them in cell culture as well as in vivo [118]. The above-mentioned findings suggest that hyperammonemia during NASH and cirrhosis may itself promote fibrosis. Ammonia is also generated from amino acids in the gut by bacteria [119]. Therefore, the composition of the gut microbiota contributes to the circulating ammonia levels. However, the exact amount of ammonia produced by gut microbiota and their role in determining serum ammonia levels in NASH and cirrhosis are not well known.

There is accumulating evidence that NAFLD is related to suboptimal liver function, even during the early stages of the disease [120,121,122,123]. Urea synthesis occurs exclusively in the liver, which is the primary location for waste nitrogen, i.e., ammonia elimination, by converting excess amino-nitrogen to urea [124]. This cycle is significantly impaired in patients with cirrhosis due to the loss of function of the hepatic cells, resulting in the accumulation of ammonia [124,125]. However, this cycle appears to be impaired even in the pre-cirrhotic stages of NAFLD. It has been documented—both in animal models as well as in humans with NAFLD—that there is a decrease in the ability for urea synthesis, together with the expression and function of urea cycle enzymes, even at the stage of simple steatosis without fibrosis (F0), which leads to a reduced ammonia elimination and thus hyperammonemia even at a non-cirrhotic stage [117,118,126,127]. Furthermore, dietary intervention resulted in the restoration of the normal urea cycle enzyme activity in animal models of NASH, which was further related to a significant reduction in liver fat content [116,128]. There is mounting evidence arguing that the presence of steatosis has a detrimental effect on mitochondrial liver function, including the urea cycle, while in vitro studies have demonstrated that the accumulation of lipids in hepatocytes leads to reduced expression of the urea cycle enzymes and thus hyperammonemia [116,118,129,130]. Therefore, steatosis in early NAFLD seems to be the cause of dysfunction in the urea cycle rather than just a coincidental finding [131]. Ammonia is a neurotoxic molecule that easily crosses the blood brain barrier and plays a key role in the pathogenesis of hepatic encephalopathy [132]. However, the pathogenesis of hepatic encephalopathy is complicated, and whilst ammonia undoubtedly plays a key role, it is not solely responsible for the neurocognitive dysfunction in hepatic encephalopathy [133]. Many studies have documented that hepatic encephalopathy manifestations may be exacerbated in an inflammatory milieu [134,135,136]. Therefore, systemic inflammation acts together with the dysfunctional nitrogen metabolism in patients with progressive liver dysfunction. It is widely accepted that hepatic encephalopathy represents a primary gliopathy, which results from astrocyte swelling and oxidative stress. Even in the absence of clinically overt hepatic encephalopathy, low-grade astrocyte swelling, which may be present in NAFLD, could impair the crosstalk between neurons and swollen astrocytes [137,138,139]. Neuroinflammation is now considered a well-established feature of hepatic encephalopathy [140,141,142,143,144,145]. It is suggested that neuroinflammation caused by hyperammonemia could be reversible by decreasing systemic ammonia levels or by anti-inflammatory treatment [146].

## 4. Therapeutic Interventions Related to Gut Microbiota for NAFLD

Patients with NAFLD present higher overall morbidity and mortality, attributed mainly to hepatic complications, cardiovascular diseases, and cancer [8]. The therapeutic management of NAFLD is focused on hepatic disease, mainly fibrosis, which is a significant prognostic factor, while also targeting metabolic comorbidities such as obesity, T2DM, and dyslipidemia. The most efficient and significant step for the therapeutic management of NAFLD, recommended by all medical societies, is lifestyle modification through a healthy diet and regular physical activity resulting in weight loss. Interestingly, intensive lifestyle modification ameliorated NAFLD in lean and overweight/obese patients [147]. However, lifestyle modifications may be difficult to achieve and maintain. Drug therapy should be directed to patients with NAFLD presented with advanced disease, i.e., NASH with fibrosis, taking into account safety issues and NAFLD comorbidities. Although many therapeutic approaches (anti-obesity, hypoglycemic, lipid lowering drugs, etc.) have been proposed and the treatment of NAFLD is a hot topic of research, there are currently no approved drugs for this indication.

### 4.1. Diet

A low-fat, low-carbohydrate diet with a caloric restriction (500–1000 kcal/day deficit to provoke a weight reduction of 0.5–1.0 kg/week) is generally recommended. Steatosis may be ameliorated by a 5% body weight loss, while a 7–10% is required for improving inflammation and fibrosis [148].

High-fat and high-cholesterol diets have been linked to hepatic steatosis, inflammation, and liver fibrosis [149]. Chronic administration of HFD in rodents has been associated with relatively increased numbers of Firmicutes and decreased numbers of Bacterioidetes species, resulting in a higher F/B ratio [150]. On the other hand, a high-fiber diet has been protective against hepatic inflammation and has been linked to the increased presence of *Akkermansia muciniphila* [151].

Fructose has been associated with dyslipidemia and insulin resistance. A study comparing fructose and glucose consumption has documented increased visceral adipose tissue volume as well as hepatic de novo lipogenesis, which has been approximately 3-fold higher in patients with NAFLD [152,153,154]. Glycotoxin levels, also known as advanced glycation end-products (AGEs), are very high in baked and fried food, especially under high-temperature cooking [155,156]. Although there is a scarcity of studies regarding AGEs and NAFLD, an animal study has reported that a high-AGEs diet provoked liver steatosis and fibrosis [157]. On the contrary, caffeine consumption has been demonstrated to be protective against the development of NAFLD as well as the progression of the stage of liver fibrosis [158]. The potential mechanism of the action of caffeine is the decrease in fat accumulation, liver inflammation, and oxidative stress as caffeine may increase glutathione and serve as an ROS scavenger at the cellular level [159]. It is also involved in restoring the F/B ratio [160]. In addition, another plausible mechanism of caffeine action is the up-regulation of the expression of Aquaporin-8 in the proximal colon, which is associated with increased growth of the *Bifidobacterium* species [161]. Moreover, coffee consumption has been shown to correlate with an amelioration of liver enzymes and a decreased risk of cirrhosis and HCC, as well as mortality rates, in a dose-dependent manner [162,163,164]. Besides, increased coffee intake has been associated with alterations in the gut microbiota composition; in particular, increased levels of *Bacteroides*, *Prevotella*, and *Porphyromonas* have been reported among high consumers of coffee [162,163,164].

Green tea extract in conjunction with epigallocatechin-3-gallate has been associated with improvements in several obesity parameters, presumably due to the restoration of the F/B ratio and the *Bacteroides*-to-*Prevotella* ratio [165,166,167]. It is noteworthy that patients with NAFLD receiving green tea extract together with 2.5% caffeine have shown remarkable improvements in liver enzymes after 12 weeks of administration [168]. Another study has confirmed the abovementioned results by reporting improvements in liver enzymes together with a significant decrease in the proportion of body fat in NAFLD patients [169]. In addition, green tea liquid consumption has also been documented to alter the human gut microbiome in many studies [169].

Omega-3 polyunsaturated fatty acids (PUFAs) are suggested to be another promising dietary agent in combating NAFLD. Omega-3 PUFAs regulate the peroxisome proliferator activated receptors (PPARs) and decrease proinflammatory cytokines as well as oxidative stress. In a recent meta-analysis, it has been suggested that omega-3 PUFAs could result in improvements in liver enzymes and the lipid profile [170]. Foods rich in omega-3 PUFAs are found in excess in the Mediterranean diet, which is well known for its beneficial effects in preventing obesity and cardiovascular diseases [170,171].

### 4.2. Exercise

Exercise, both aerobic and resistance training, is one of the most promising lifestyle modifications and has been proven to prevent or even reverse NAFLD/NASH [172,173,174]. As physical inactivity has been associated with the progression of the disease, exercise is suggested to be beneficial not only because of the weight loss but, above and beyond this, also because of the metabolic benefits which accompany exercise training [175,176]. In particular, exercise has been implicated in the increased production of several angiogenic factors, thus favoring fatty acid utilization and leading to a decreased entry of FFA into the liver [175,176,177,178]. Furthermore, alterations in the composition of the gut microbiota have been reported in response to exercise training. More specifically, in animal models, exercise has been associated with a reduction in the presence of *Parabacteroides*, *Flavobacterium*, and *Alkaliphilus* [179]. Besides, exercise has been related to a relatively increased presence of Verrucomicrobia and a reduction in Proteobacteria in overweight women, as well as a relative reduction in the F/B ratio in patients with T2DM [180]. Alterations that have been associated with exercise also include a reduced presence of the mouse-associated Bacteroidales S24-7 and Rikenellaceae families, which have already been related to increased intrahepatic TG content in mouse models [181]. The significant effect of exercise in NAFLD patients could be partially explained by the interplay between the gut and the liver. For example, exercise has been proven to increase the abundance of SCFAs, mainly butyrate [182]. In addition, exercise exerts a beneficial effect on other gut-derived metabolites related to hepatic metabolism, such as BAs and choline. Furthermore, exercise has been documented to exert epigenetic changes by modulating the expression of various genes implicated in lipid metabolism, such as *SREBP-1c*, *FAT/Cd36*, and *C/EBPa* [175,176,177,178]. Exercise provides more beneficial effects than diet alone. Notably, in a study comparing exercise to calorie restriction in HFD-fed animal models, exercise has been shown to increase insulin sensitivity and result in greater LDL reduction, mainly on account of the exercise-induced microbiome alterations [180]. Such modifications in the gut microbiota have been related to improvements in serum LDL cholesterol, liver fat mass, and liver TG [180]. Notably, Barton et al. have documented an increase in the abundance of *Akkermansia* spp. among athletes with a more active exercise training regimen than among more sedentary individuals, while Allen et al. have reported an increase in the presence of *Faecalibacterium* spp. with exercise among lean individuals when compared to patients with obesity [183,184]. Interestingly, both *Akkermansia* and *Faecalibacterium* are known for their beneficial effects on health [185].

### 4.3. Bariatric Surgery

Bariatric surgery is a successful therapy for severe obesity and related metabolic disorders, leading to permanent weight loss and amelioration of metabolic and inflammatory comorbidities such as NAFLD [186,187]. The most widely performed types of bariatric surgery are laparoscopic Roux-en-Y gastric bypass (RYGB) and laparoscopic sleeve gastrectomy (SG). After bariatric surgery, the anatomy, physiology, and energy and substrate metabolism adapt to a new and dynamic status [188,189]. The specific mechanisms through which bariatric surgery affects the composition of the gut microbiota remain to be elucidated. However, several factors have been postulated as being involved in the alteration of the gut microbiota, such as changes in dietary habits, gastrointestinal anatomy, nutrients as well as gastrointestinal transit time and pH, and bile acid metabolism [189]. Studies have documented that after bariatric surgery there is a mild increase in microbial gene richness but not to the levels seen in lean humans [7,190]. There are numerous studies which have investigated the influence of RYGB on the gut microbiota. For example, in a study of 16 patients who underwent RYGB, the gut microbiome was analyzed before and 3 months following surgery. Before surgery, there was relative abundance in the phyla Firmicutes and Actinobacteria, whereas Verrucomicrobia were less abundant when compared to the lean controls. After RYGB, the abundance of these phyla was approximately similar to that in the healthy controls. Only Proteobacteria abundance was more prominent after RYGB and lower in lean subjects [191]. At the genus level, *Blautia*, *Roseburia*, *Faecalibacterium* (Firmicutes), and *Bifidobacterium* (Actinobacteria) were reduced. It was noteworthy that these genera were still more abundant when compared to lean subjects [184,185]. At the species level, *Streptococcus* spp., *Akkermansia muciniphila* (Verrucomicrobia), *Roseburia feces*, *Roseburia hominis*, and *Enterococcus faecalis* were in relative abundance, whereas *Faecalibacterium prausnitzii* was reduced after RYGB [192,193,194,195,196,197,198,199]. Other studies analyzing the composition of the gut microbiota after SG have documented an abundance of the species *C. comes*, *D. longicatena*, *Clostridiales bacterium*, *Anaerotruncus colihominis*, *Akkermansia muciniphila*, and *B. thetaiotaomicron* [191,192,193,194,195,196,197,198]. Moreover, the abundance of the species *Akkermansia muciniphila*, *Roseburia* spp., Bacteroidetes, and *Bifidobacterium* have been demonstrated to be increased. One year after SG, the abundance of the phylum Actinobacteria was increased compared to the baseline and three months postoperatively [199]. Overall, several studies have reported substantial alterations in the composition of the gut microbiota after bariatric surgery. Different interventions have resulted in differential microbial profiles, whilst only partial restoration towards the lean gut microbiota composition was noted. Different methods of bariatric surgery, patient characteristics, differences in sample sizes and methodologies as well as the existence of comorbidities, such as T2DM, could account for the abovementioned variable results [192,193,194,195,196,197,198,199].

Bariatric surgery leads to prolonged weight loss with subsequent reductions in hepatic fat mass, inflammation, and fibrosis [186,187,188]. Indeed, a meta-analysis of 43 studies with 2809 participants has reported a dose-response association with liver inflammation, ballooning, and NAFLD/NASH resolution [186,187,188,189]. There is ongoing interest and research on this issue and the results of the ongoing trials are eagerly anticipated.

### 4.4. Probiotics

As gut dysbiosis appears to contribute to the pathogenesis of NAFLD, probiotics have been tested in the prevention and treatment of NAFLD. The term “probiotics” first appeared in 1974 but has since evolved to its current definition as “live microorganisms that confer a health benefit when consumed in adequate amounts,” as proposed by the World Health Organization (WHO) in 2002 [200]. To date, probiotics have been shown to improve the lipid profile as well as liver function tests of patients with NAFLD; however, data regarding hepatic histologic alterations are inconclusive. Trials with the administration of probiotics have been confounded by dietary factors which affect intestinal microbiota composition, different formulas, and dosages.

*Bifidobacterium* and *Lactobacillus* strains are still the most widely used probiotics in functional foods and dietary supplements, but next-generation probiotics, such as *Faecalibacterium prausnitzii*, *Akkermansia muciniphila*, or *Clostridia* strains, have demonstrated promising results [201]. VSL#3 and modified VSL#3 are mixtures of probiotics of the genera *Lactobacillus*, *Bifidobacterium*, and *Streptococcus* or *Lactobacillus* alone. VSL#3 has been documented to have a protective effect against NAFLD by inhibiting inflammatory pathways, such as c-Jun N-terminal kinase (JNK) and NF-κB, and restoring the number of natural killer (NK) T cells in the liver, caused by high-fat feeding [202,203]. On the contrary, in another animal model study, whilst VSL#3 did not have any effects on methionine-choline-supplemented diet-induced hepatic steatosis or inflammation, it improved liver fibrosis by down-regulating TGF-β signaling [204]. Other studies in rodents have found that probiotics improved the gut microbiota composition by maintaining tight junctions, thus restoring the intestinal mucosal barrier and suppressing the serum LPS levels. Liver inflammatory markers as well as serum cytokines levels were reported to be decreased, in parallel with a reduction in the serum LPS and liver TLR-4 mRNA concentrations [204,205]. In addition, *Lactobacillus plantarum* NA136 has been shown to reduce the mass of fat tissues in HFD-fed and fructose-fed animal models of NAFLD and to decrease serum liver enzymes. Besides, *L. plantarum* NA136 reduced lipogenesis and enhanced fatty acid oxidation by stimulating the AMPK pathway to phosphorylate ACC and suppress SREBP-1/FAS signaling in a NASH animal model. Moreover, *L. plantarum* NA136 reduced oxidative stress in the liver by stimulating the AMPK/NF-E2-related factor 2 (Nrf2) signaling in a NAFLD animal model. These results point towards a promising role of *L. plantarum* NA136 in improving NAFLD [204,205]. *Lactobacillus paracasei* has been found to reduce the expression of TLR-4, CCL2, and TNF-α and improve liver steatosis. In particular, *L. paracasei* has been documented to reduce the proportion of M1 Kupffer cells and increase that of M2, leading to an M2-dominant shift in the liver in a NASH animal model, a shift similar to that ascertained in animal models of obesity [204,205]. The addition of a combination of three probiotics has resulted in an enhanced production of lactate concentrations, which in turn led to the activation of lactate-consuming bacteria growth, resulting in a significant enhancement in SCFAs production. Furthermore, the abovementioned combination of probiotics has shown immunomodulatory potential, including enhanced production of anti-inflammatory cytokines, i.e., IL-10 and IL-6, and reduction in the production of proinflammatory chemokines, i.e., IL-8, CXCL 10, and Monocyte Chemoattractant Protein 1 (MCP)-1 [204,205]. However, the data regarding combinations of probiotics are difficult to interpret, as different dosages and proportions of different probiotics are used, while there are other factors to be taken in account, such as the duration of treatment as well as the timing of data collection after the discontinuation of treatment.

An animal study has found that *Faecalibacterium prausnitzii* may reduce adipose tissue inflammation in rodents and improve the metabolic parameters of liver function [206,207]. Nevertheless, this bacterial species is very oxygen-sensitive and difficult in cultivation and preservation; more research to improve its preservation and viability with the addition of antioxidants, such as riboflavin, cysteine, and cryoprotectant inulin, is ongoing. Another potential probiotic is *Akkermansia muciniphila*, a mucin-degrading bacterium, which has been found to reduce fat mass and ameliorate dyslipidemia in animal models [208]. Several meta-analyses have also advocated the use of probiotics in NAFLD, documenting improvements in various metabolic and inflammatory parameters. The combined use of *Akkermansia muciniphila* with metformin has been shown to induce better results than monotherapy alone with regard to serum levels of liver enzymes and sonographic characteristics [208,209,210,211,212,213,214].

### 4.5. Prebiotics

Prebiotics constitute non-digestible carbohydrates, which lead to significant changes in the composition or activity of the gut microbiota, resulting in beneficial effects on host health [200]. Most data refer mainly to two chemical substances, inulin-type fructans and galactooligosaccharides. These data have reported increases in the production of *Bifidobacteria* and *Lactobacilli*, thus conferring beneficial alterations in the composition and activity of the gut microbiota [200]. Prebiotics may confer benefits to both NAFLD and NASH. In rodents, prebiotics have been documented to induce changes in the composition of the gut microbiota, resulting in increased plasma glucagon-like peptide-2 (GLP-2) levels, thus ameliorating the function of the gut barrier. Furthermore, prebiotics have been demonstrated to decrease liver inflammation and improve metabolic parameters in obesity [210]. Besides, prebiotics, mainly inulin and oligofructose, have been found to better control the growth of *Faecalibacterium prausnitzii* and *Bifidobacterium* as well as decrease serum LPS concentrations, due to increased GLP-1 production and the GLP-2-mediated trophic effect on the integrity of the gut barrier. Among patients with obesity, supplementation with oligofructose induced weight loss by regulating appetite hormones such as ghrelin and peptide YY [215].

### 4.6. Synbiotics

Synbiotics are a combination of prebiotics and probiotics, which may be used to replace a dysfunctional gut microbiota [200]. A meta-analysis of probiotics, prebiotics, and synbiotics supplementation in NAFLD has confirmed significant decreases in the body mass index as well as improvements in liver enzymes. Probiotics/synbiotics use has been related to a significant decrease in alanine aminotransferase (ALT) levels and in liver stiffness assessed by elastography [215].

### 4.7. Antibiotics

Antibiotics have also been suggested as one of the therapeutic options in NAFLD. Nevertheless, there is conflicting evidence about the efficacy of antimicrobial treatments. The use of norfloxacin and neomycin has been shown to improve liver function in cirrhotic patients by causing changes in bacterial translocation and overgrowth [216]. On the contrary, another study has not shown any beneficial effects on liver function in NAFLD patients administered with norfloxacin. Gangarapu et al. also showed that rifaximin treatment significantly reduced proinflammatory cytokines, ALT, and the NAFLD-liver fat score. This improvement by antibiotics was attributed to alterations in the gut microbiota population and bile acid metabolism as well as to reduced FXR signaling and decreased ceramide levels in the liver [217]. Hence, there is no established role for antibiotics in NAFLD, although longitudinal large-scale studies are lacking. The risk for bacterial drug resistance and potential side effects should also not be overlooked [185].

### 4.8. FMT

FMT has not been very widely used, apart from special circumstances, such as the treatment of severe/recurrent *Clostridioides difficile* infection, and to a lesser extent in inflammatory bowel disease, metabolic syndrome, and hepatic encephalopathy [141,218]. There is a scarcity of studies regarding the use of FMT in NAFLD. Only recently, a study in animal models has documented a beneficial effect on lipid accumulation in the liver as well as liver histology after FMT [219]. Nowadays, FMT has been used successfully in cirrhotic patients with hepatic encephalopathy and severe alcoholic hepatitis [219,220]. However, the success rates of FMT are mainly dependent on the “donor” characteristics, in particular fecal microbiota richness, diversity, and compatibility [221,222,223]. Further studies are needed to assess the long-term efficacy together with the safety of FMT in patients with NAFLD.

Overall, weight loss of 7–10% of total body weight in patients with biopsy-proven NASH remains the cornerstone of the treatment of NAFLD/NASH. Diet, physical exercise, and bariatric surgery have yielded impressive results, while the use of other conventional measures, such as vitamin E supplementation, the addition of pioglitazone—which may cause weight gain but has been documented to decrease inflammation and liver fibrosis—or glucagon like peptide (GLP) analogs seem to exert hepato-protective properties [8].

### 4.9. Bacteriophages

As most bacteriophages and archaeal viruses are specific to bacterial and archaeal strains, respectively, they may be used to target dysbiotic parts of the microbiota in patients with metabolic disorders in the near future [224]. The therapeutic strategy of targeting a single strain, specifically cytolytisin-positive *E. faecalis*, with the advent of a bacteriophage has only recently been found to be effective in ethanol-induced liver disease experiments using humanized mice [224]. More specifically, patients with alcoholic hepatitis have increased numbers of *E. faecalis* in their feces. The presence of this cytolytic *E. faecalis* has been shown to relate to the severity of liver disease as well as mortality among these patients. Using humanized mice that were colonized with bacteria from the feces of patients with alcoholic hepatitis, Duan et al. embarked on investigating the therapeutic effects of bacteriophages that target cytolytic *E. faecalis*. They have documented that these bacteriophages decrease cytolysin in the liver and mitigate ethanol-induced liver disease in humanized mice. This method with the advent of bacteriophages raises the possibility of modulating the gut microbiota [225,226,227]. Clinical trials are further required to assess the relevance of the abovementioned findings in humans and to test whether this therapeutic approach is effective for patients with NAFLD/NASH.

## 5. Limitations of the Studies

Although there is a plethora of studies advocating the key role of gut microbiota in maintaining homeostasis and preventing gut dysbiosis and the development of NAFLD via the gut–liver axis, there is significant heterogeneity in terms of basic and clinical research on NAFLD/NASH. This heterogeneity could be attributed to the variety and differential degrees in lifestyle modifications, such as diet, the intensity and duration of exercise training, and the administration or not of pro/pre/synbiotics. In addition, there is a paucity of data regarding the role of gut microbiota in NAFLD in humans, and there is no sufficient evidence on the potential role of therapeutics, either in the form of nutritional agents, such as caffeine and polyphenols, or in the form of pro/pre/synbiotics. The reasons behind this scarcity of studies in humans are complicated; NAFLD itself is a complex clinical entity, not always biopsy-proven and with different fibrosis severity, ranging from F0 to F4.

In addition, pro/pre/synbiotics may be administered in different formulas with different concentrations and combinations. Overall, large randomized controlled trials (RCTs) are mandatory with the use of current advances in metagenomics techniques. Multi-omics, although costly and often difficult to perform and interpret, are necessary in studying the role of the gut microbiome in the pathogenesis of NAFLD/NASH [228].

## 6. Perspectives and Conclusions

Nowadays, NAFLD has become a pandemic attributed mainly to Western diet, obesity, and a mostly sedentary lifestyle. Although NAFLD is much more common than in the past, the current methods of diagnosis still have limitations as they are invasive (liver biopsy) or have a low predictive value (noninvasive biomarkers). Human biology should not overlook the gut microbiota, which produce or modulate various chemicals and trigger host reactions, thereby affecting multiple functions, including immunity and metabolism. In this review, we have highlighted the distinct microbiota profile in patients with NAFLD/NASH, which may be correlated to the severity and progression of cirrhosis or HCC. Nevertheless, gut microbiota composition may vary between population groups and different stages of NAFLD, making any conclusive or causative claims about the gut microbiota profile in NAFLD patients challenging. In conclusion, the current limitations of the therapeutic strategies in the fight against NAFLD/NASH should prompt scientists to shed light on newer approaches, with the advent of modern technology, and explore more options regarding the interplay between the gut microbiota, its metabolites, and NAFLD/NASH in terms of diagnosis, prognosis, and therapeutics. It seems likely that modulations in the gut microbiota in patients suffering from liver diseases will be very promising in the near future. Large-scale RCTs in humans are required to evaluate the beneficial properties of probiotics, prebiotics, and synbiotics, their ideal dose, the duration of supplementation, and the durability of their beneficial effects as well as their safety profile in the prevention and treatment of NAFLD.

## Figures and Tables

**Figure 1 biomolecules-12-00056-f001:**
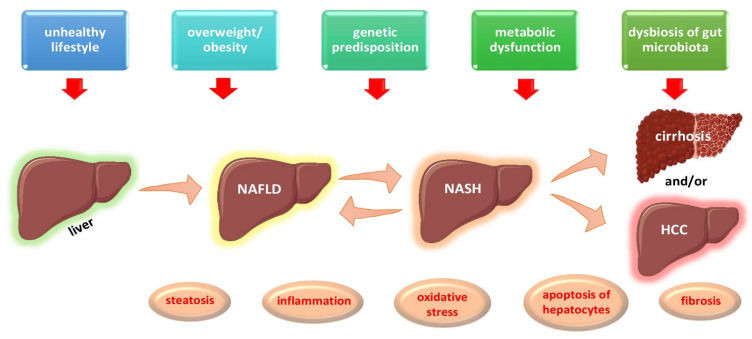
Multi-hit etiopathogenetic model of NAFLD progression and staging. Abbreviations: HCC, Hepatocellular Carcinoma; NAFLD, Non-Alcoholic Fatty Liver Disease; NASH, Non-Alcoholic Steatohepatitis. (All images are originated from the free medical website http://smart.servier.com/ by Servier licensed under a Creative Commons Attribution 3.0 Unported License, accessed on 1 October 2021).

**Figure 2 biomolecules-12-00056-f002:**
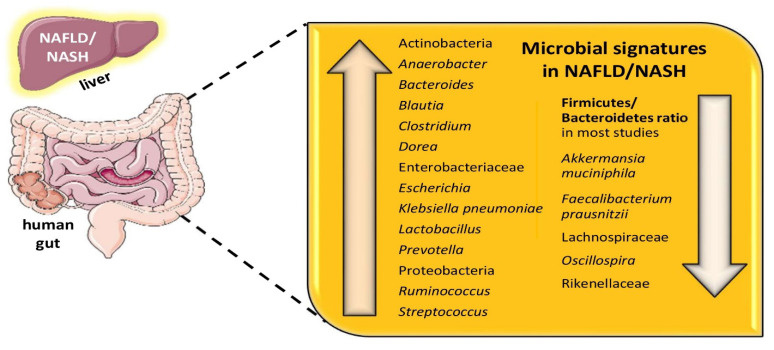
Gut microbial signatures found in NAFLD/NASH. Abbreviations: NAFLD, Non-Alcoholic Fatty Liver Disease; NASH, Non-Alcoholic Steatohepatitis. (All images are originated from the free medical website http://smart.servier.com/ by Servier licensed under a Creative Commons Attribution 3.0 Unported License, accessed on 1 October 2021).

**Figure 3 biomolecules-12-00056-f003:**
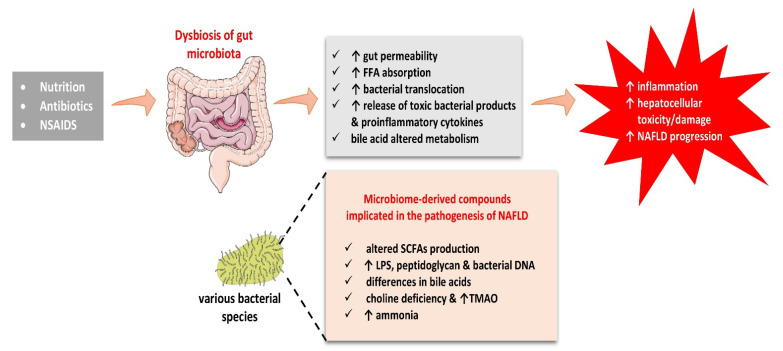
Dysbiosis of gut microbiota may explain the inflammatory process and hepatotoxicity of microbiome-derived compounds implicated in the pathogenesis of NAFLD. Abbreviations: FFA, Free Fatty Acids; LPS, Lipopolysaccharide; NAFLD, Non-Alcoholic Fatty Liver Disease; NSAIDs, Non-Steroid Anti-Inflammatory Drugs; SCFAs, Short-chain Fatty Acids; TMAO, Trimethylamine N-oxide. (All images are originated from the free medical website http://smart.servier.com/ by Servier licensed under a Creative Commons Attribution 3.0 Unported License, accessed on 1 October 2021).

## Data Availability

Not applicable.

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
