# Peer review of "Understanding the Role of the Gut Microbiome and Microbial Metabolites in Non-Alcoholic Fatty Liver Disease: Current Evidence and Perspectives"

_biomolecules, 2021, doi:10.3390/biom12010056_

Round 1

Reviewer 1 Report

Review of Vallianou et al. 2021. “Understanding the role of the gut microbiome and microbial metabolites in non-alcoholic fatty liver disease: current evidence and perspectives”

This is a review of the role of the gut microbiome in NAFLD across a large number of different areas related to disease. Overall, the review is reasonable and covers a complex area of research. My major comments are shown below:

General comments

  1. In many areas, the description of the results of microbiome analyses are not worded quite right. Authors need to be careful when discussing microbiome results, since these analyses are most often showing changes in relative abundance, not absolute abundance. Please be careful and accurate when reporting analysis results. Authors should also indicate whether analyses were performed with 16S ribosomal RNA gene amplicon sequencing or shotgun metagenome sequencing. These have different biases and that may also lead to differences in results.
  2. Also, please note that only species and genus names are italicized
  3. Also, please note that the F/B ratio is a bit tricky – as it both may be highly affected by method of sequencing (e.g., different primer sets, shotgun vs 16S, etc) – and also a bit fuzzy as there are huge diversities of organisms within each of these phyla, and the overall application of F/B can be a bit crude.
  4. Discussions of indirect effects of factors such as diet are shown (e.g., caffeine leads to a decrease in fat accumulation) – but are there any direct effects on microbial communities?
  5. Not much discussion of the actual activity of the organisms identified. There could be more discussion of the likely physiological activity of the affected taxa (e.g., SCFA-producers).

Editorial comments

L35: “an abundance in Bacteroidetes” – strange wording here

L36: Are these data generated using 16S? Also what is this relative too?

L55: Use “approximately” for 25%

L64: The overall prevalence of NASH is approximately between 1.5-6.5%.  AMONG what group?

L71: “Supposed to haunt” – very strange wording

L102: What are these genes?

L104: Do you really mean epigenetic factors? These seem like environmental factors?

L110: Replace “microbiome” with “metagenome”.

L114: “less frequently encountered” – these organisms are still highly frequently encountered in mammalian fecal microbiomes, but at lower relative abundance relative to Firmicutes and Bacteroidetes.

L117: Again – these do not seem to be epigenetic factors

L123: “Seem to prevail” – strange wording

L136: I think translocation may be more

L168: Which E.coli (other than engineered strains) product ethanol?

L180-181: This last sentence doesn’t seem to flow with the whole paragraph

L202: Not all the mentioned taxa are species

L205-210: I think the important distinction is between 16S rRNA gene amplicon sequencing and shotgun metagenome sequencing. Also, sampling location is important – e.g., feces vs cecum vs sigmoid biopsy, etc.

L351: Do you mean choline-enriched? Seems confusing.

L353: What about methanogens from TMA? What are the broader implications of TMAO? There are associations with cardiovascular issues.

L386: Do you mean restoration of the normal urea cycle?

L451-457: Studies shown direct green tea effects on human microbiome (e.g., Yuan, Xiaojie, et al. "Green tea liquid consumption alters the human intestinal and oral microbiome." Molecular nutrition & food research 62.12 (2018): 1800178.)

L479: Note the S24-7 is really only a mouse-associated organism

L599: I don’t think “non-viable food compounds” is the correct term. Usually, “nondigestible carbohydrates” or similar.

Reviewer 2 Report

The manuscript entitled “Understanding the role of the gut microbiome and microbial metabolites in non-alcoholic fatty liver disease: current evidence and perspectives” is a comprehensive review that summarizes some of the findings concerning the role of gut dysbiosis in the development and progression of non-alcoholic fatty liver disease (NAFLD) and therapeutic interventions related to gut microbiota for this disease. In my opinion this paper fits the aims and scope of the journal. NAFLD is a very important topic because its currently represents the most common etiology of chronic liver disease in children and adults world-wide. It is a well written high-quality manuscript. The title and abstract are informative and they give a clear idea of what to expect from the article.The collected data based on the currently published literature can expand our understanding of the role of gut microbiome in NAFLD pathogenesis with special regards to various metabolites (e.g. SFA, LPS/endotoxin, bile acids, choline, TMAO, and ammonia) as well as gut microbial signature in NAFLD and provide insight into potential preventive and therapeutic intervention related to gut microbiota (e.g. probiotics, prebiotics, synbiotics or FMT). The authors cited and commented the most important articles concerning this topic. It is important that authors pointed the limitations of published studies concerning gut microbiome role and NAFLD. Therefore, I strongly agree with authors that “muli-omics are necessary in studying the role of gut microbiome in the pathogenesis of NAFLD/NASH” and “large-scale RCT-s in humans are required to evaluate the beneficial properties” of potential drugs in NAFLD therapy.  The paper is scientifically accurate, clearly written and does not leave questions unanswered. The overall rating of this manuscript is high. This manuscript will be interesting for scientists as well as clinicians. In my opinion this article is ready for publication.

Reviewer 3 Report

In the current review entitled "Understanding the Role of the Gut Microbiome and Microbial 2 Metabolites in Non-alcoholic Fatty Liver Disease: Current Evi-3 dence and Perspectives" authors aimed to summarize the role of the gut microbiome 46 and metabolites in NAFLD pathogenesis and to discuss potential preventive and therapeutic interventions related to gut microbiome. The current review is well designed and well written, however there are some comments to be considered before publications:

Major points:

1- There are several published reviews describing the same topic. The authors should indicate in the introduction what it is novel in this review in comparison to the previously published ones. Examples:

1- Gut microbiota and their metabolites in the progression of non-alcoholic fatty liver disease

2- Disease, Drugs and Dysbiosis: Understanding Microbial Signatures in Metabolic Disease and Medical Interventions

3- Gut microbiome and microbial metabolites: a new system affecting metabolic disorders

2- Why is the design of table 1 and table 2 different? These tables should describe the gut microbiota signatures in NAFLD and therefore should represent more or less similar kind of data. Also, "remarks" column in table 2 includes unnecessary details.

3- Figure 3 is entitled "Dysbiosis of gut microbiota and microbial metabolites implicated in the pathogenesis of NAFLD". However, it does not include any detailed data regarding dysbiosis of gut microbiota. It includes only details about microbial metabolites. This figure can be improved to be self-representative.

4- Subtitles mentioned in section 3 are different from compound names mentioned in figure 3. The name of microbiome derived compounds mentioned in the figure should be written in the same manner in the subtitles for each compound. Example: in figure, the author wrote LPS while the subtitle they wrote endotoxin and in text they wrote LPS-endotoxin. Also, the order of compounds in the figure should be in the same order of subsections and even the first two compounds should be present in details but not completely absent as in case of ethanol and butanone.

5- The authors mention differences between certain gut micorbiome species in details only in the last section regarding therapeutic intervention while the other sections include few or no details about the difference between gut microbiome species regarding the topic of the section. The second aim of this review as mentioned by authors is to shed light on the distinct microorganisms that seem to prevail in NAFLD and their metabolic signatures but this aim was poorly described.

Minor points:

Certain abbreviations are mentioned in full name more than one time such as TMAO. Please, check and revise the other abbreviations.

Round 2

Reviewer 3 Report

The authors did all the required improvements.